# Understanding of Collective Atom Phase Control in Modified Photon Echoes for a Near-Perfect Storage Time-Extended Quantum Memory

**DOI:** 10.3390/e22080900

**Published:** 2020-08-15

**Authors:** Rahmat Ullah, Byoung S. Ham

**Affiliations:** 1Center for Photon Information Processing, and School of Electrical Engineering and Computer Science, Gwangju Institute of Science and Technology, Gwangju 61005, Korea; rahmatktk@comsats.edu.pk; 2Quantum Optics Laboratory, Department of Physics, COMSATS University, Islamabad 44000, Pakistan

**Keywords:** quantum memory, photon echoes, coherence control, rephrasing, Maxwell-Bloch

## Abstract

A near-perfect storage time-extended photon echo-based quantum memory protocol has been analyzed by solving the Maxwell–Bloch equations for a backward scheme in a three-level system. The backward photon echo scheme is combined with a controlled coherence conversion process via controlled Rabi flopping to a third state, where the control Rabi flopping collectively shifts the phase of the ensemble coherence. The propagation direction of photon echoes is coherently determined by the phase-matching condition between the data (quantum) and the control (classical) pulses. Herein, we discuss the classical controllability of a quantum state for both phase and propagation direction by manipulating the control pulses in both single and double rephasing photon echo schemes of a three-level system. Compared with the well-understood uses of two-level photon echoes, the Maxwell–Bloch equations for a three-level system have a critical limitation regarding the phase change when interacting with an arbitrary control pulse area.

## 1. Introduction

Over the last decade, modified photon echo techniques have been intensively studied and applied to quantum memory applications to overcome the fundamental limitation of population inversion in conventional photon echoes, for which the population inversion excited by an optical π pulse results in quantum noises caused by spontaneous and/or stimulated emissions [1,2,3,4,5,6,7,8,9,10,11,12,13,14,15,16,17]. Photon echo has also been experimentally studied in a highly doped crystal Tm:YAG [18] and in ruby upon excitation through an optical fiber [19]. Although some of these techniques have been successful for quantum state storage and retrieval [8,9,10,11,12,13], the understanding of collective atom phase control is still limited, where absorptive photon echoes have been involved in controlled atomic frequency comb (AFC) echoes [8,9] and doubly rephased (DR) photon echoes [10,11,12,13]. To coherently manipulate the absorptive photon echoes, a controlled coherence conversion (CCC) process has been proposed [20] and experimentally demonstrated [21]. Recent observations of photon echoes in single [8,9] and double [10,11,12,13] rephasing photon echo schemes contradict the CCC theory but reveal coherence leakage by Gaussian light pulses, resulting in echo generations all the time, regardless of the pulse area, whose maximum efficiency reaches as high as 26% [22]. In addition to our previous studies on numerical [14,15,16] and analytical [17] approaches, herein, we discuss the CCC theory by using the commonly applied Maxwell–Bloch (MB) approach to correct the critical mistakes in previous works [8,9,10,11,12,13] and thus to contribute to the implementation of photon echo–based quantum memory.

In the study of modified photon echo-based quantum memories, using MB equations has been a common theoretical tool. The MB equations have the advantage of using both space and time variables and thus are practical for calculations of photon echo retrieval efficiency, with respect to the optical depth (or physical length) of an ensemble medium [1,2,3,4]. However, the MB approach is unable to give exact solutions of individual atom phase evolutions, and thus continuous tracing of ensemble coherence change in the time domain is impossible. This limitation requires us to totally rely upon inevitable assumptions for the atom-field interactions, such as rephasing and CCC. In other words, the MB theory prevents us from obtaining exact answers to the phase change of the ensemble coherence, with respect to the pulse area of an interacting optical field. Here, this critical mistake means that the MB approach never reveals the induced π− phase shift [20] by the π−π control pulse sequence in a three-level system [8,9]. Please note that the coherence must be distinguished in between a single atom qubit and a macroscopic ensemble qubit, where the sign of coherence has nothing to do with the single qubit measurement, but is critical to the ensemble qubit due to macroscopic coherent transients. In other words, the π−π control pulse-induced negative sign in ensemble coherence, which is the same as the data pulse, can never be radiated out of the medium regardless of population inversion [20,22]. Here, the single atom has nothing to do with coherent transients, where the sign of coherence does not matter either.

To overcome this limitation of the MB approach, we have dealt with full numerical [14,15,16] and analytical [17] solutions based on time-dependent density matrix equations to exactly trace the coherence evolutions of an ensemble. As a result, we have introduced the controlled double rephasing (CDR) echo protocol [14] based on CCC [20]. Although CDR is perfect for investigating temporal coherence behaviors of coherent transient phenomena, such techniques of numerical and analytical calculations are limited in optical depth-dependent photon echo efficiency. In the present report, we comply with the MB equations, firstly to confirm the CDR echo protocol with previous results of CCC in references [17] and [20], secondly to analyze the critical mistakes in references [8,9,10,11,12,13], due to incorrect assumptions of π−π control pulse sequence [8,9] and the misunderstanding of the absorptive echo [10,11,12,13], and finally to discuss ensemble phase evolutions and their phase control in both single [8,9] and double rephasing schemes [10,11,12,13,14]. Near-perfect retrieval efficiency in quantum memories is critical for both fault-tolerant quantum computing [23] and loophole-free quantum cryptography [24]. Thus, the present report of the near-perfect storage time-extended quantum memory protocol should shed light on future quantum information research using quantum memories.

### 1.1. Theory: Maxwell–Bloch Equations

#### 1.1.1. A. Conventional Two-Pulse Photon Echo

We considered a three-level optical ensemble medium composed of *N* indistinguishable atoms. The energy level diagram and pulse sequence of the present CDR echo scheme are depicted in Figure 1. The data (D), the first rephasing (R_1_) and the second rephasing (R_2_) pulses were resonant to the transition of |1〉−|2〉 to satisfy the requirements of the DR photon echo scheme [10,11,12,13,14,15,16,17]. The control pulses, C_1_ and C_2_, were resonant between states |2〉 and |3〉 to enable CCC [14,15,16,17,18,19]. Initially, all atoms of the medium were in the ground state, |1〉. For an ideal system, all decay rates were neglected, unless specified otherwise. All light pulses were collinear (or near collinear) in the *z*-axis. To make the first echo (E_1_) silent to avoid affecting the final echo (E_2_), both rephasing pulse propagation directions were set to be opposite (backward) to that of the D pulse [10]. To satisfy the backward photon echo condition, the control pulses were set to counter-propagate each other [19]: ${\vec{k}}_{E_{2}}={\vec{k}}_{C_{1}}+{\vec{k}}_{C_{2}}-{\vec{k}}_{D}$; each pulse *j* was characterized by a wave vector ${\vec{k}}_{j}$. Unlike phase mismatching in silent echo (E_1_) formation [10], which is determined by D and R_1_, the final echo (E_2_) propagation direction (${\vec{k}}_{E_{2}}$) is determined by the control pulses [14]. The MB equations for the atomic coherence and the D pulse were respectively denoted, as follows:(1)$$\frac{\partial }{\partial t}\sigma_{12}\left(z,t,\Delta \right)\;=\;i\Delta \sigma_{12}\left(z,t,\Delta \right)+i\varepsilon_{D}\left(z,t\right),$$
(2)$$\frac{\partial }{\partial z}\varepsilon_{D}\left(z,t,\Delta \right)\;=\;\frac{i\alpha }{2\pi }\int_{-\infty }^{\infty }\sigma_{12}\left(z,t,\Delta \right)d\Delta ,$$
where $\sigma_{12}\;=\;|1\rangle \langle 2|$ represents the atomic coherence, ε_D_ is the data pulse, α is the optical depth parameter, also known as the attenuation coefficient with unit *m*^−1^, and Δ is the detuning of the atom (see Appendix A). The ensemble was inhomogeneously broadened by $\Delta^{\prime }=\;\sum_{j}\Delta_{j}$. Here, we considered the case of a detuned atom for simplicity. Thus, the solution for atomic coherence is given by:(3)$$\sigma_{12}\left(z,t,\Delta \right)\;=\;i\;\int_{-\infty }^{t}\varepsilon_{D}\left(z,\overset{´}{t}\right)e^{i\Delta (t-\overset{´}{t})}d\overset{´}{t}.$$

The positive sign of this equation represents that the atomic coherence excited by the D pulse is absorptive. It should be noted that $\sigma_{12}\left(z,t,\Delta \right)\;=\;-\rho_{12}\left(z,t,\Delta \right)$, where $\rho_{12}\left(z,t,\Delta \right)$ is the density matrix element. The solution of Equation (2) gives the well-known Beer’s law: $\varepsilon_{D}\left(z,t\right)\;=\;\varepsilon_{D}\left(0,t\right)e^{-\alpha z/2}$. The D pulse is assumed to be fully absorbed by the medium, thereby transferring its quantum information (phase, amplitude, polarization, etc.) into the collective atomic coherence. We assumed that the D pulse was too weak to treat it as a quantum state. Thus, the ground state population change arising from the D pulse excitation was neglected: $\sigma_{11}\left(z,t,\Delta \right)\;=\;1\;\mathrm{and}\;\sigma_{22}\left(z,t,\Delta \right)\;=\;0$.

With the time delay of T, as shown in Figure 1b, the rephasing pulse (R_1_) arrives at $t\;=\;t_{R_{1}}.$ The R_1_ pulse has a π pulse area, and its propagation direction is opposite to that of the D pulse (${\vec{k}}_{R_{1}}\;=\;-{\vec{k}}_{D}$). Here, the function of the R_1_ pulse is to swap the populations between states |1〉 and |2〉: $\sigma_{11}\left(z,t_{D},\Delta \right)\overset{R_{1}}{↔}\sigma_{22}\left(z,t_{R_{1}},\Delta \right)$. With the weak D pulse, the swapping result is $\sigma_{22}\left(z,t,\Delta \right)=1\;\mathrm{and}\;\sigma_{11}\left(z,t,\Delta \right)=0$. Thus, the atomic coherence arising from the application of the R_1_ pulse becomes:(4)$$\frac{\partial }{\partial t}\sigma_{12}\left(z,t,\Delta \right)\;=\;i\Delta \sigma_{12}\left(z,t\right)-i\varepsilon_{R_{1}}\left(z,t\right)$$

The solution of Equation (4) is:(5)$$\sigma_{12}\left(z,t,\Delta \right)\;=\;e^{i\Delta \left(t-t_{R_{1}}\right)\;}\sigma_{12}\left(z,t_{R_{1}},\Delta \right)-i\;\int_{t_{R_{1}}}^{t}\varepsilon_{R_{1}}\left(z,\overset{´}{t}\right)e^{i\Delta (t-\overset{´}{t})}d\overset{´}{t}.$$

Because the π pulse reverts the coherence evolution direction, $\sigma_{12}\left(z,t_{R_{1}},\Delta \right)$ is equal to the conjugate of Equation (3) at $t\;=\;t_{R_{1}}$, namely $\sigma_{12}\left(z,t_{R_{1}},\Delta \right)\;=\;{\left[\sigma_{12}\left(z,t\;=\;t_{R_{1}},\Delta \right)\right]}^{†}$. Therefore, Equation (5) can be rewritten as:(6)$$\sigma_{12}\left(z,t,\Delta \right)\;=-\;i\;e^{-i\Delta \left(2t_{R_{1}}-t\right)}\int_{-\infty }^{\infty }\varepsilon_{D}^{†}\left(z,\overset{´}{t}\right)e^{i\Delta \overset{´}{t}}d\overset{´}{t}-i\;\int_{t_{R_{1}}}^{t}\varepsilon_{R_{1}}\left(z,\overset{´}{t}\right)e^{i\Delta (t-\overset{´}{t})}d\overset{´}{t},$$
where the first term represents free evolution and the second term represents interaction with the rephasing pulse. The ensemble coherence generated by the D pulse became in phase and resulted in a photon echo E_1_ at $t\;=\;t_{E_{1}}\;=\;2t_{R_{1}}-t_{D}$. The propagation direction of the first echo, E_1_, determined by the first rephasing pulse, R_1_, was ${\vec{k}}_{E_{1}}\;=\;2{\vec{k}}_{R_{1}}-{\vec{k}}_{D}\;=\;-3{\vec{k}}_{D}$. Due to the phase mismatch between D and E_1_, the echo signal (macroscopic coherence) could not be generated due to complete out of phase [10]: silent echo.

#### 1.1.2. B. DR Echo

To rephase the system one more time, the second rephasing (R_2_) π pulse was followed by E_1_ to satisfy the requirements of the DR photon echo scheme. In the DR scheme, the excited state population after the final echo, E_2_, was the same as that after the D pulse excitation. This means that all excited-state atoms in the DR scheme should contribute to the echo signal without contributing to quantum noises. The MB equation for DR was similar to that for D, and the optical coherence solution for the final echo, E_2_, is:(7)$$\sigma_{12}\left(z,t,\Delta \right)\;=\;e^{i\Delta \left(t-t_{R_{2}}\right)\;}\sigma_{12}\left(z,t_{R_{2}},\Delta \right)+i\int_{t_{R_{2}}}^{t}\varepsilon_{R_{2}}\left(z,\overset{´}{t}\right)e^{i\Delta (t-\overset{´}{t})}d\overset{´}{t},$$
where $\sigma_{12}\left(z,t_{R_{2}},\Delta \right)$ is equal to the conjugate of Equation (6) at $t\;=\;t_{R_{2}}$:(8)$$\begin{array}{c} \sigma_{12}\left(z,t,\Delta \right)=i\;e^{-i\Delta \left(2t_{R_{2}}-2t_{R_{1}}-t\right)\;}\int_{-\infty }^{\infty }\varepsilon_{D}\left(z,\overset{´}{t}\right)e^{-i\Delta \overset{´}{t}}d\overset{´}{t}+\\ i\;e^{-2i\Delta t_{R_{2}}\;}\int_{-\infty }^{\infty }\varepsilon_{t_{R_{1}}}^{†}\left(z,\overset{´}{t}\right)e^{i\Delta (t+\overset{´}{t)}}d\overset{´}{t}+i\;\int_{t_{R_{2}}}^{t}\varepsilon_{R_{2}}\left(z,\overset{´}{t}\right)e^{i\Delta (t-\overset{´}{t})}d\overset{´}{t}. \end{array}$$

The final echo, E_2_, was formed at $t\;=\;t_{E_{2}}$ as the rephasing result of E_1_ by R_2_, and its propagation direction was forward if ${\vec{k}}_{R_{2}}\;=\;{\vec{k}}_{R_{1}}$: ${\vec{k}}_{E_{2}}\;=\;2{\vec{k}}_{R_{2}}-{\vec{k}}_{E_{1}}\;=\;2{\vec{k}}_{R_{2}}-\left(2{\vec{k}}_{R_{1}}-{\vec{k}}_{D}\right)\;=\;{\vec{k}}_{D}$. However, the retrieved signal, E_2_, was absorbed by the medium due to absorptive coherence, as shown in Equation (8). In other words, echo E_2_ could not be radiated from the medium. This fact has already been explored numerically [14,15,16] and analytically [17]. From now on, we discuss and correct critical mistakes in previous analyses [8,9,10,11,12,13]. The observation of DR echoes [10,11,12,13] has been understood as a coherence leakage due to an imperfect rephasing process by Gaussian-distributed light in a transverse mode, resulting in the leakage-caused maximum retrieval efficiency at 26% [22].

## 2. Discussion

### 2.1. A. CDR Echo

To fix the absorptive echo problem in DR, the CDR echo protocol was proposed, whereby the control pulse pair C_1_ and C_2_ were added, as shown in Figure 1b [14]. As discussed in references [14,15,16,17,20], the function of the control pulses is not only to convert the coherence between optical and spin states via population transfer, but also to induce a collective phase shift. Unlike the DR scheme of Equation (8), the propagation vector of E_2_ can be controlled to be backward if ${\vec{k}}_{C_{1}}\;=\;-{\vec{k}}_{C_{2}}$: ${\vec{k}}_{E_{2}}\;=\;{\vec{k}}_{C_{1}}+{\vec{k}}_{C_{2}}-{\vec{k}}_{D}\;=\;-{\vec{k}}_{D}$ [14,21]. Here, the rephasing pulses had nothing to do with the phase matching for ${\vec{k}}_{E_{2}}$. Unlike the forward echo in the conventional two-pulse photon echo scheme, which suffers from reabsorption by noninteracting atoms, according to Beer’s law, the backward echo E_2_ is free from reabsorption, resulting in near-perfect echo efficiency [1,2,21]. To eliminate any potential two-photon coherence between states |1〉 and |3〉, the C_1_ pulse was delayed by *δ*T from R_1_. Here, it should be noted that the macroscopic two-photon coherence was sustained only within the overall optical coherence time determined by the inverse of the atom broadening $\Delta^{\prime }$. However, individual atom coherence was sustained regardless of atom broadening for the optical phase decay time T_2_, where $T_{2}\gg \Delta T$ [20]. Here, the D pulse duration was practically comparable to (or a bit longer than) $1/\Delta^{\prime }$. Thus, by simply neglecting δT, the atomic coherence at $t_{C_{1}}$ can be expressed as:(9)$$\sigma_{12}\left(z,t_{C_{1}},\Delta \right)\;=\;i\;e^{-i\Delta \left(2t_{R_{2}}-2t_{R_{1}}-t_{C_{1}}\right)\;}\int_{-\infty }^{\infty }\varepsilon_{D}\left(z,\overset{´}{t}\right)e^{-i\Delta \overset{´}{t}}d\overset{´}{t}.$$

In Equation (9), we have only considered the first term of Equation (8), which is related to the free evolution of the coherences generated by the D pulse. The second and third terms are associated with the rephasing fields. Because the first echo, E_1_, is silent, and the second echo, E_2_, is not emitted, owing to its absorptive coherence, we can remove the last two terms of Equation (8). With a π C_1_ pulse, optical and spin coherence, respectively, satisfy the following relations: $\sigma_{12}\left(z,t,\Delta \right)\;=\;\cos\left(\frac{\pi }{2}\right)\sigma_{12}\left(z,t_{C_{1}},\Delta \right)\;=\;0$; $\sigma_{13}\left(z,t,\Delta \right)\;=\;e^{i\pi /2}\sigma_{12}\left(z,t_{C_{1}},\Delta \right)$ (see references. [14,15,16,17,20]). The C_1_ pulse induced a π/2 phase shift and locked the optical coherence until C_2_ arrived. Here, the novel property of the π/2 phase shift by the π pulse area of C_1_ (or C_2_) originated in the Raman (two-photon) coherence, where a 2π Raman pulse in a three-level system actually played as a π pulse does in a two-level system: see Figures 3 and 4 of reference [25] and Figure 4 of reference [26]. There is no way to expect this novel property from the MB approach. Without a priori understanding of the coherence behavior in a three-level system, the same mistake has been repeated in the controlled AFC echoes [8,9]; this will be discussed further in section B.

The atomic coherence after the C_2_ pulse is:(10)$$\frac{\partial }{\partial t}\sigma_{12}\left(z,t,\Delta \right)\;=\;i\Delta \sigma_{12}\left(z,t,\Delta \right).$$

Equation (10) is obtained by substituting $\varepsilon_{l}\;=\;0$ and $\sigma_{13}\left(z,t,\Delta \right)\;=\;0$ (no spin coherence after C_2_) into Appendix A. The solution of Equation (10) is:(11)$$\sigma_{12}\left(z,t,\Delta \right)\;=\;\sigma_{12}\left(z,t_{C_{2}},\Delta \right)e^{i\Delta \left(t-t_{C_{2}}\right)\;}.$$

The C_2_ pulse also brings a π/2 phase shift, while swapping the spin and optical coherence:(12)$$\begin{array}{cc} \sigma_{12}\left(z,t_{C_{2}},\Delta \right)\;= & e^{i\pi /2}\sigma_{13}\left(z,t,\Delta \right)\;=\;e^{i\pi }\sigma_{12}\left(z,t_{C_{1}},\Delta \right)\\ & \;=\;-i\;e^{-i\Delta \left(2t_{R_{2}}-2t_{R_{1}}-t_{C_{1}}\right)\;}\int_{-\infty }^{\infty }\varepsilon_{D}\left(z,\overset{´}{t}\right)e^{-i\Delta \overset{´}{t}}d\overset{´}{t} \end{array}$$

Substituting Equation (12) into Equation (11) gives:(13)$$\sigma_{12}\left(z,t,\Delta \right)\;=-i\;e^{-i\Delta \left(t_{C_{2}}-t_{C_{1}}+2t_{R_{2}}-2t_{R_{1}}-t\right)\;}\int_{-\infty }^{\infty }\varepsilon_{D}\left(z,\overset{´}{t}\right)e^{-i\Delta \overset{´}{t}}d\overset{´}{t},$$
where the negative sign represents that the echo E_2_ is emissive due to the π phase shift induced by the control pulse pair, which has already been extensively explored numerically [14,15,16] and analytically [17]. Thus, the echo E_2_ propagates backward without reabsorption and is radiated out of the medium with near-perfect retrieval efficiency; this will be discussed further in Figure 2.

### 2.2. A. Single Rephased Photon Echo

Now we consider the coherence evolution for a controlled single rephasing scheme with R_2_ = 0 in Figure 1 [20,21]. This scheme itself cannot be used for quantum memory because the population inversion has not been solved yet. However, this scheme itself implies the AFC echo scheme, whereby all excited-state atoms freely decay into a dump state. The AFC scheme can be easily obtained by swapping the pulse sequence of D and R_1_ and substituting R_1_ with a repeated weak two-pulse train (see Appendix A). Therefore, by neglecting the population constraint issue without R_2_, the final optical coherence could be obtained by applying the control pulses to Equation (6), as follows (see Appendix A):(14)$$\sigma_{12}\left(z,t,\Delta \right)\;=i\;e^{-i\Delta \left(t_{C_{2}}-t_{C_{1}}+2t_{R_{1}}-t\right)\;}\int_{-\infty }^{\infty }\varepsilon_{D}^{†}\left(z,\overset{´}{t}\right)e^{i\Delta \overset{´}{t}}d\overset{´}{t}.$$

The positive sign of this equation represents the absorptive coherence, which is the same problem as in the DR scheme in Equation (8). In other words, the π–π control pulse sequence added to the single rephasing scheme inverts the system coherence to be absorptive [20]. To convert the absorptive echo into an emissive echo, simply one more control Rabi flopping was added, for which the C_2_ pulse area was increased to 3π, i.e., $\sigma_{12}\left(z,t,\Delta \right)\overset{C_{1}\left(\pi \right)}{\to }e^{-\left(\frac{\pi }{2}\right)i}\sigma_{13}\left(z,t,\Delta \right)\overset{C_{2}\left(\pi \right)}{\to }e^{\pi i}\sigma_{12}\left(z,t,\Delta \right)\overset{C_{2}\left(\pi \right)}{\to }e^{-\left(\frac{3\pi }{2}\right)i}\sigma_{13}\left(z,t,\Delta \right)\overset{C_{2}\left(\pi \right)}{\to }e^{2\pi i}\sigma_{12}\left(z,t,\Delta \right)$, and thus $\sigma_{12}\left(z,t,\Delta \right)$ was negative. This means that the π–π control pulse sequence in references [8,9] must be corrected to be a π–3π control pulse sequence. The π–π control pulse sequence in reference [1] uses a Doppler effect–caused π-phase shift, resulting in an emissive echo. This is not the case for a solid medium [2,14,20]. However, the observation of controlled AFC echoes with a π–π control pulse set [8,9] is due to the imperfect rephasing by commercial light sources with Gaussian spatial distributions [22]. The coherence leakage in a DR scheme induced by the Gaussian pulse limits the echo efficiency to ~10% on average, regardless of the rephasing pulse area, whereas its maximum efficiency reaches 26% for a π/2 pulse area in a DR scheme [22].

### 2.3. A. Near Perfect Retrieval Efficiency in CDR

We now calculate the echo efficiency of the present CDR echo scheme shown in Figure 1. Because the echo E_2_ was emitted in the backward direction, the atomic coherence and optical field in the backward direction could be respectively represented, as follows [27]:(15)$$\frac{\partial }{\partial t}{\left[\sigma_{12}\left(z,t,\Delta \right)\right]}_{b}=i\Delta {\left[\sigma_{12}\left(z,t,\Delta \right)\right]}_{b}+i\varepsilon_{b}\left(z,t\right),$$
(16)$$\frac{\partial }{\partial z}\varepsilon_{b}\left(z,t\right)=-\frac{i\alpha }{2\pi }\int_{-\infty }^{\infty }{\left[\sigma_{12}\left(z,t,\Delta \right)\right]}_{b}d\Delta .$$

The solution of Equation (15) is:(17)$${\left[\sigma_{12}\left(z,t,\Delta \right)\right]}_{b}={\left[\sigma_{12}\left(z,0,\Delta \right)\right]}_{b}e^{i\Delta t}+i\;\int_{0}^{t}\varepsilon_{b}\left(z,\overset{´}{t}\right)e^{i\Delta (t-\overset{´}{t})}d\overset{´}{t}$$
where ${\left[\sigma_{12}\left(z,0,\Delta \right)\right]}_{b}$ is obtained by setting *t* = 0 in Equation (13).
(18)$${\left[\sigma_{12}\left(z,0,\Delta \right)\right]}_{b}=\sigma_{12}\left(z,t=0,\Delta \right)=\;-i\;e^{-i\Delta \left(t_{C_{2}}-t_{C_{1}}+2t_{R_{2}}-2t_{R_{1}}\right)\;}\int_{-\infty }^{\infty }\varepsilon_{D}\left(z,\overset{´}{t}\right)e^{-i\Delta \overset{´}{t}}d\overset{´}{t}$$

Inserting Equation (18) into Equation (17) gives:(19)$$\begin{array}{c} {\left[\sigma_{12}\left(z,t,\Delta \right)\right]}_{b}=-i\;e^{-i\Delta \left(t_{C_{2}}-t_{C_{1}}+2t_{R_{2}}-2t_{R_{1}}-t\right)\;}\int_{-\infty }^{\infty }\varepsilon_{D}\left(z,\overset{´}{t}\right)e^{-i\Delta \overset{´}{t}}d\overset{´}{t}\\ +i\;\int_{0}^{t}\varepsilon_{b}\left(z,\overset{´}{t}\right)e^{i\Delta (t-\overset{´}{t})}d\overset{´}{t} \end{array}$$

Equations (19) and (16) are solved by means of Laplace transformation. For the backward scheme, ε_b_ (*L*, ω) = 0 because there is no field at *z* = *L*. Assuming an ideal case of complete absorption at *z* = *L*, solves Equations (19) and (16) yields:(20)$$\varepsilon_{b}\left(0,\omega \right)=\;\left(1-e^{-\alpha L}\right)e^{-i\omega \left(t_{C_{2}}-t_{C_{1}}+2t_{R_{2}}-2t_{R_{1}}-t\right)\;}\varepsilon_{D}\left(0,\omega \right)$$

The inverse Fourier transform gives:(21)$$\varepsilon_{b}\left(0,t\right)=\left(1-e^{-\alpha L}\right)\varepsilon_{D}\left(0,t-(t_{C_{2}}-t_{C_{1}}+2(t_{R_{2}}-t_{R_{1}}))\right)$$
where the echo is emitted at $t=t_{E_{2}}=t_{C_{2}}-t_{C_{1}}+2\left(t_{R_{2}}-t_{R_{1}}\right)+t_{D}$. Because $t_{C_{2}}-t_{C_{1}}$ can be lengthened to be much longer than $t_{R_{1}}-t_{D}$ in a Zeeman scheme [28], an additional but important function of the control pulse set is storage time extension [8,9,21]. The efficiency of the final backward echo E_2_ is given by:(22)$$\eta ={\left(1-e^{-\alpha L}\right)}^{2}$$
where η can be the near unity in an optically dense medium, as shown in Figure 2.

On the other hand, the forward echo E_2_ can be obtained by setting ${\vec{k}}_{C_{1}}={\vec{k}}_{C_{2}}$, where ${\vec{k}}_{C_{1}}={\vec{k}}_{D}$: ${\vec{k}}_{E_{2}}={\vec{k}}_{C_{1}}+{\vec{k}}_{C_{2}}-{\vec{k}}_{D}=2{\vec{k}}_{C_{1}}-{\vec{k}}_{D}={\vec{k}}_{D}$. Here, the difference frequency between C_1_ and D is just ~10 MHz, which is about 10^−8^ times the frequency of D in a rare-earth doped solid. The atomic coherence and the forward field satisfy the following relations:(23)$$\frac{\partial }{\partial t}{\left[\sigma_{12}\left(z,t,\Delta \right)\right]}_{f}=i\Delta {\left[\sigma_{12}\left(z,t,\Delta \right)\right]}_{f}+i\varepsilon_{f}\left(z,t\right)$$
(24)$$\frac{\partial }{\partial z}\varepsilon_{f}\left(z,t\right)=\frac{i\alpha }{2\pi }\int_{-\infty }^{\infty }{\left[\sigma_{12}\left(z,t,\Delta \right)\right]}_{f}d\Delta$$

By following the same procedure as used for the backward echo above, the optical field propagating in the forward direction can be expressed as follows:(25)$$\varepsilon_{f}\left(L,t\right)=\alpha Le^{-\frac{\alpha L}{2}}\varepsilon_{D}\left(0,t-(t_{C_{2}}-t_{C_{1}}+2(t_{R_{2}}-t_{R_{1}}))\right)$$

Thus, the retrieval efficiency of the forward echo E_2_ is (*αL*)^2^e^−αL^ (see the dotted curve in Figure 2). The forward echo E_2_ suffers from reabsorption.

In Figure 2, we plot the echo efficiency of both the forward (dotted) and backward (solid) echo schemes for Figure 1. For the forward echo, the retrieval efficiency reaches 54% at maximum and then exponentially decreases due to the reabsorption process. On the other hand, the echo efficiency for the backward scheme approaches unity for *αL* ≫ 0, as expected, according to reference [1] and as demonstrated in reference [21].

## 3. Conclusions

We analyzed and discussed the modified photon echo schemes of controlled double rephasing (CDR) for near-perfect echo efficiency with no population inversion. We also pointed out the absorptive echo problems in previously demonstrated single (AFC) and double rephasing photon echo schemes, where the mistake in the controlled AFC echoes arises from a wrong assumption applied to the Maxwell–Bloch approach for a three-level system. Compared with a two-level system, whose coherence harmonic is 2π-based, the three-level system shows 4π-based coherence harmonics. The Maxwell–Bloch method never gives an exact solution to the phase change of the ensemble coherence in its interaction with an arbitrary optical pulse area, unless known a priori, as in the two-level system. With the help of full numerical [14,15,16] and analytic [17] calculations using time-dependent density matrix equations, our present Maxwell–Bloch calculation results confirmed the CDR echo theory. Thus, the present report gives a clear understanding of collective atom phase control in modified photon echoes for quantum memory applications. The controlled coherence conversion process via control Rabi flopping to a third state was also investigated as a means of coherence inversion in a solid medium, whereby the absorptive coherence of the photon echo in the double rephasing scheme was converted into an emissive echo. Finally, we discussed the near-perfect CDR echo efficiency in a backward scheme using counter-propagating control Rabi pulses.

## Figures and Tables

**Figure 1 entropy-22-00900-f001:**
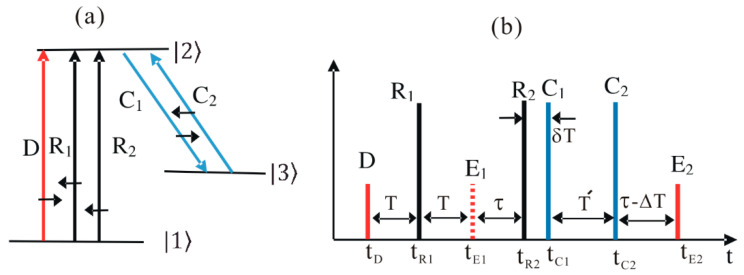
Schematics of controlled double rephasing (CDR) echoes. (**a**) Energy level diagram for the CDR echo. The short black arrows indicate the pulse propagation direction. (**b**) Pulse sequence for (**a**), where t_j_ is the arrival time of pulse j.

**Figure 2 entropy-22-00900-f002:**
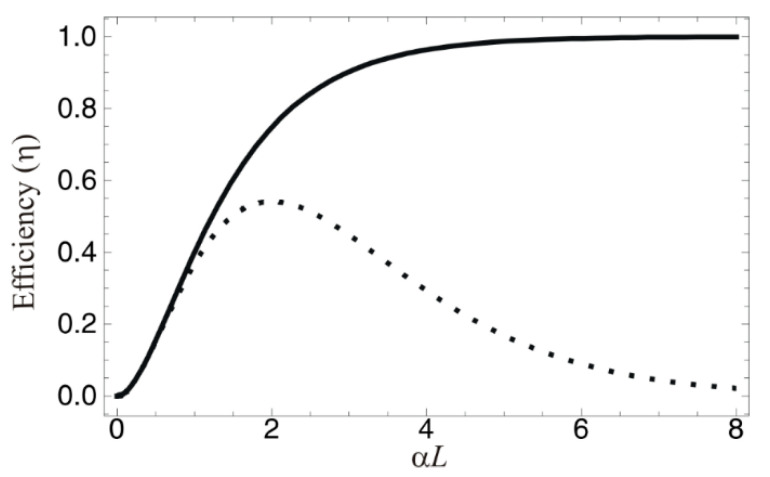
Efficiency of CDR echoes versus optical depth αL. The solid curve is for the backward final echo (E_2_): see Equation (22). The dotted curve is for the forward echo E_2_: (*αL*)^2^e^−αL^.

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
