# Peer review of "Understanding of Collective Atom Phase Control in Modified Photon Echoes for a Near-Perfect Storage Time-Extended Quantum Memory"

_entropy, 2020, doi:10.3390/e22080900_

Round 1

Reviewer 1 Report

The authors analyzed and discussed the modified photon echo schemes of controlled double rephrasing for near perfect echo efficiency with no population inversion. They also pointed out the absorptive echo problems in previously demonstrated single (AFC) and double rephrasing photon echo schemes, where the mistake in the controlled AFC echoes arises from a wrong assumption applied to the Maxwell–Bloch approach for a three-level system. However, the paper is not innovative enough to be published in Entropy. Moreover, the authors should pay more attention to writing. The paper can be submitted to another journal. Thus, I cannot recommend this manuscript for publication in Entropy.

Author Response

Reply:

The novelty of the present paper is to indicate the problems of refs. [8-13] which are not correct in terms of Maxwell-Bloch (MB) approach, where MB lacks in coherence phase. For this, we have made detail calculations for the controlled echo using the   control-pulse sequence, where AFC echoes [8,9] are not correct as mentioned in terms of absorptive echo coherence [9-13]. Such a wrong result was analyzed carefully in ref. 20. This is why controlled AFC echoes based on Fig. 1 cannot even generate a 10% memory efficiency. Unless more than 50% echo efficiency, it cannot be applied for quantum memories. Although the same echo efficiency has been calculated for a backward photon echo in ref. 25 (PRA 75, 032327), it does not based on the correct coherence as answered to Referee 3 for Comment 3. Our calculations in Fig. 2 for the echo efficiency is first trial for the control double rephrasing scheme of Fig. 1. By the way, the submitted manuscript was already proof read by an English native speaker.

Reviewer 2 Report

The paper describes in detail the theory of controlled coherence conversion using Maxwell–Bloch approach which allows one to describe correctly the coherence behavior in a three-level system, namely the controlled atomic frequency comb echoes and doubly rephased photon echoes. The goal of the paper is to solve the absorptive echo problem. The authors made accurate calculations to achieve high retrieval efficiency in a controlled doubly rephased photon echo scheme.

Presented information can be useful for quantum optics and quantum information research community. However, the paper contains a few points, which reduce the generally positive impression of the article. Generally, describing photon echo schemes, the authors do not pay any attention to the papers concerning the experiments on backward photon echo and photon echo-spectroscopy of high-{alpha}L-materials. Some research papers in this field can be included in the article (e.g., Samartsev et al., Incoherent backward photon echo in ruby upon excitation through an optical fiber, Laser Physics Letters. – 2007. – V. 4, # 7. – P. 534-537 and Kalachev A.A. et al., Optical echo-spectroscopy of highly doped Tm:YAG, Laser Physics Letters. – 2008. – V. 5, # 12. – P. 882-886).

Author Response

Reply:

Thank you for your valuable comments, positive reviewing, and reference recommendations. We have added two references of the mentioned above into refs. 18 and 19.

Reviewer 3 Report

The manuscript is devoted to the actively studied topic of quantum optics and quantum information, namely optical quantum memory. The authors consider multipulse photon-echo-based-quantum memory protocols in a three-level system using the Maxwell-Bloch equations and conclude that the collective atomic phase control is crucial for understanding quantum memory functioning and achieving near perfect retrieval efficiency.

Issues:

  • The authors write in the introduction that in the present work they analyze the critical mistakes in [8-13] due to incorrect assumptions. It is not clear which assumptions were made in that works. In addition, it is written that “There is no way to expect this novel property from MB approach” (line 152). I don’t understand from the text why MB equations in three-level systems do not describe the coherence evolution and phase relations properly. The atomic coherence is considered as a function of coordinate and frequency, which is equivalent to considering the atoms individually. Finally, the main results of the present study shown in Fig. 2 do not contradict with the previous theory so that no critical mistakes could be seen.
  • The authors use the terms ‘absorptive coherence’ and ‘emissive coherence’ when discussing Eqs. (3) and (13), respectively. The type of the coherence is determined by the sign of the r.h.s. in these equations. However, it also depends on the relative phase between the atomic coherence and field amplitude which is not fixed in these equations. In particular, if the phase of the input pulse is suddenly shifted by pi, the type of coherence is also changed. I think the terms ‘absorptive coherence’ and ‘emissive coherence’ should be defined considering this point as well.
  • All the variables used in Eqs. (1) and (2) should be defined. In addition, the definition of inhomogeneous width Delta’ as a sum of the atomic detunings Delta_j looks strange because in this case it becomes very large. There should be some average value.

I propose considerable revision of the manuscript.

Author Response

Referee 3

The manuscript is devoted to the actively studied topic of quantum optics and quantum information, namely optical quantum memory. The authors consider multipulse photon-echo-based-quantum memory protocols in a three-level system using the Maxwell-Bloch equations and conclude that the collective atomic phase control is crucial for understanding quantum memory functioning and achieving near perfect retrieval efficiency.

Comment 1:

The authors write in the introduction that in the present work they analyze the critical mistakes in [8-13] due to incorrect assumptions. It is not clear which assumptions were made in that works.

Reply 1:

We have slightly modified the above mentioned sentence (line 58 and 59 in revised manuscript) to clearly point out the mistakes and incorrect assumption in refs. [8-13]. In Refs. [8-9], the assumption of the pi-pi control pulse leads to absorptive echo. Likewise, refs. [10-13] based on double rephased (DR) photon echo scheme also result in the same absorptive echo problem. We have studied in detailed these mistakes in the Discussion section, where the absorptive echo is discussed in section “DR echo” (see line 113 in the revised manuscript). The incorrect assumption with a pi-pi control pulse sequence is also discussed in detailed in “Single rephased photon echo” (line 167 in the revised manuscript). The pi-pi control pulse sequence must be correct with pi-3pi control pulse sequence ( also see lines 182 and 183 in the revised manuscript). Please refer that the technical problem of Gaussian light was represented in ref. 22 which will be published in Entropy.

Comment 2:

In addition, it is written that “There is no way to expect this novel property from MB approach” (line 152). I don’t understand from the text why MB equations in three-level systems do not describe the coherence evolution and phase relations properly. The atomic coherence is considered as a function of coordinate and frequency, which is equivalent to considering the atoms individually.

Reply 2:

Thank you for your valuable comments regarding the MB approach. When we deal with MB equations to study photon echo, we must do some assumptions, e.g., after the D pulse in Fig. 1, such that atoms are in the ground state even though coherence is generated with some atoms excited. The equations for coherence in DR (equations 8) and in CDR (equation 13) do not give the exact analytical solutions at all. For a correct analysis, we must solve these equations numerically. This is how exact solutions of coherence in terms of pulse area are not possible. In lines 151-155, we have tried to explain that we cannot obtain the total phase shift of pi induced by the control pulses C1 and C1 in the MB approach. No one can get such information of this phase shift from equation 13. The pi shift can be observed by a full numerical approach [14-16, 20] or by full analytical [17] solutions based on time-dependent density matrix equations.       

Comment 3:

Finally, the main results of the present study shown in Fig. 2 do not contradict with the previous theory so that no critical mistakes could be seen.

Reply 3:

Thank you again for your valuable comments. As efficiency is the ratio of absolute square of output pulse to input pulse, i.,e , |E_out/E_in|^2, where input pulse (D-pulse) is related to the coherence (equation 2), therefore theoretical  efficiency should be same for both positive coherence (absorptive) and negative coherence (emissive). The experimental observations, however, cannot reach the theoretical expectation due to the absorptive echo: Please see there is no observations of echo efficiency of even 10% for the controlled echo scheme of ref. [8] based on the pi-pi control-pulse sequence. In a practical point of view, any pulse scheme can generate a certain level of echo efficiency due to imperfectness of pulse shaping with commercial Gaussian lights [22], e.g., a maximum 26% retrieval efficiency, as we have mentioned in lines 35-38 and 127-128 in the revised manuscript.

Comment 4:

The authors use the terms ‘absorptive coherence’ and ‘emissive coherence’ when discussing Eqs. (3) and (13), respectively. The type of the coherence is determined by the sign of the r.h.s. in these equations. However, it also depends on the relative phase between the atomic coherence and field amplitude which is not fixed in these equations. In particular, if the phase of the input pulse is suddenly shifted by pi, the type of coherence is also changed. I think the terms ‘absorptive coherence’ and ‘emissive coherence’ should be defined considering this point as well.

Reply 4:

We agree with referee that absorptive and emissive coherence can also be discussed with relative phase. In this manuscript, our main objective is to analyze the behavior of the atomic coherence with respect to pulse area. Because of the AOM controlled pulse area, the atomic coherence is dominated by a slow process and thus experimentally controlled by the AOM phase rather than the given optical phase. If we studied the effect of relative phase in coherence, then we would not be able to show the mistakes and incorrect assumption in the AFC echo schemes. We feel that it is a good problem to study the behavior of atomic coherence with respect to the relative phase and definitely in future we will look for it. 

Comment 5:

All the variables used in Eqs. (1) and (2) should be defined. In addition, the definition of inhomogeneous width Delta’ as a sum of the atomic detunings Delta_j looks strange because in this case it becomes very large. There should be some average value.

Reply 5:

Thanks gain for your detailed comments. We have defined all the parameters in the revised manuscript (see lines 81-83). The calculations after equation (2) include the Delta for inhomogeneous broadening.

Round 2

Reviewer 1 Report

The authors have answered all the comments of reviews. I am satisfied with their answers and, hence, now I consider the work can be accepted as is.

Author Response

Reviewer 1

The authors have answered all the comments of reviews. I am satisfied with their answers and, hence, now I consider the work can be accepted as is.

Reply

We thank for your recommendation of publication in Entropy.

Reviewer 2 Report

The authors have answered all the comments of reviews. I think that the article can be accepted for publication in Entropy in the present form.

Author Response

Reviewer 2

The authors have answered all the comments of reviews. I think that the article can be accepted for publication in Entropy in the present form.

Reply

We thank for your recommendation of publication in Entropy.

Reviewer 3 Report

The authors have considered carefully my objections and gave answers and performed changes on the manuscript. I have no additional questions but I would like to make additional comments relating to the replies 2 and 3.

Reply 2:

Indeed, equations corresponding to the weak pulse excitation regime do not provide information about the phase shift induced by the control pulses. However, MB approach is not limited to the weak field regime, and, in general case, Bloch equations are equivalent to the density matrix equations. Therefore, I can agree that proper phase shifts were not taken into account in the previous works, but I still don’t agree that MB approach does not allow one to do this. The authors can consider Bloch equations instead of the density matrix equations and obtain the same results for the atomic phase shift.

Reply 3:

I still don’t understand why the mistakes in Refs. [8-13] are called critical if the total phase shift does not change neither efficiency nor fidelity of quantum memory. On the contrary, it seems that the total phase correction plays no role (cf., the total phase of a qubit state). Why the authors pay so much attention to the difference between the absorptive and emissive coherence throughout the text if their correction does not change the efficiency? The authors write (Reply 3) that the experimental observations cannot reach the theoretical expectation due to the absorptive echo. But this statement contradicts their theory.

Provided the issues above are discussed, I would recommend this manuscript for publication in Entropy.

Author Response

Reviewer 3

The authors have considered carefully my objections and gave answers and performed changes on the manuscript. I have no additional questions but I would like to make additional comments relating to the replies 2 and 3.

Comment 1

Indeed, equations corresponding to the weak pulse excitation regime do not provide information about the phase shift induced by the control pulses. However, MB approach is not limited to the weak field regime, and, in general case, Bloch equations are equivalent to the density matrix equations. Therefore, I can agree that proper phase shifts were not taken into account in the previous works, but I still don’t agree that MB approach does not allow one to do this. The authors can consider Bloch equations instead of the density matrix equations and obtain the same results for the atomic phase shift.

Reply 1

We agree that Bloch equations are equivalent to the density matrix equations. In ref. 17, we have already applied the density matrix approach to show the coherence evolutions of an ensemble in terms of pulse area. In a two-level system, the MB approach gives both coherence evolutions and phase relations properly for other phenomenon not limited to weak field approximation. However, in a three-level system of refs. 8 and 9, the pi phase shift induced to the ensemble coherence by the pi-pi control pulse sequence cannot be revealed by the MB approach as mentioned in lines 158-160: “There is no way to expect this novel property from the MB approach. Without a priori understanding of the coherence behavior in a three-level system, the same mistake has been repeated in the controlled AFC echoes [8,9]; this will be discussed further in section B.” This is the main point how people have repeatedly committed the same mistake in their theoretical and experimental studies. As already demonstrated, the echo efficiency in the controlled AFC scheme [8,9] is about one per cent, which is one order of magnitude below compared with that in a two-level system.

Comment 2

I still don’t understand why the mistakes in Refs. [8-13] are called critical if the total phase shift does not change neither efficiency nor fidelity of quantum memory. On the contrary, it seems that the total phase correction plays no role (cf., the total phase of a qubit state). Why the authors pay so much attention to the difference between the absorptive and emissive coherence throughout the text if their correction does not change the efficiency? The authors write (Reply 3) that the experimental observations cannot reach the theoretical expectation due to the absorptive echo. But this statement contradicts their theory.

Reply 2

As we have mentioned in our previous reply, the sign of ensemble coherence decides either absorption or emission which is critical in photon echoes as a macroscopic coherent transient behavior. For example, a weak data pulse-induced ensemble coherence is denoted by a negative sign revealing absorption, while the first echo is denoted by a positive sign revealing emission. In that sense a negative sign of the echo coherence cannot be emitted regardless of the population distribution. Such an induced pi phase shift by the pi-pi control pulse sequence can never be revealed or obtained from the MB approach unless you do full numerical calculations as shown in ref. 20.

For the doubly rephased echo [10-13], the second echo coherence is inverted compared with the first echo, resulting in an absorptive echo. Of course, their main interest is in the silencing the first echo via phase mismatching. To invert the absorptive coherence to an emissive one, the same control pulse sequence used in refs. [8,9] must be added [14-16]. This is why we call it a ‘critical’ mistake. Please see Fig. 5f of ref. 20 for the coherence evolution as a function of time. To help general readers, a sentence is added in lines 51-52: Here, the critical mistake means that the MB approach never reveals the induced phase shift [20] by the  control pulse sequence [8,9]. For both controlled AFC and DR echoes, the same explanation about the critical mistake is repeated in lines 58-63: “In the present Report,… ,secondly to analyze the critical mistakes in Refs. [8-13] due to incorrect assumptions of   control pulse sequence [8,9] that leads to the absorptive echo [10-13],…”

The echo observations in refs. [8-13] are not because the theory or experimental scheme is correct, but because the spatial shape of laser light is Gaussian, resulting in imperfect pulse area. Due to this imperfectness, the maximum echo efficiency observed cannot exceed 26%, and it is true as demonstrated experimentally in their papers. By the way, the previous reply of “The experimental observations, however, cannot reach the theoretical expectation due to the absorptive echo…,” means that the echo observation cannot reach more than 26% in those pulse schemes, where the 26% is due to the Gaussian pulse shape-induced error. Obviously, a quantum memory requires at least 50% retrieval efficiency to satisfy the no-cloning regime.